# Enhanced Bactericidal Effect of Calcinated Mg–Fe Layered Double Hydroxide Films Driven by the Fenton Reaction

**DOI:** 10.3390/ijms24010272

**Published:** 2022-12-23

**Authors:** Lei Chen, Yijia Yin, Linjia Jian, Xianglong Han, Xuefeng Zhao, Donghui Wang

**Affiliations:** 1School of Materials Science and Engineering, Hebei University of Technology, Tianjin 300130, China; 2State Key Laboratory of Oral Diseases & National Clinical Research Center for Oral Diseases & Department of Orthodontics, West China Hospital of Stomatology, Sichuan University, Chengdu 610041, China; 3School of Health Sciences and Biomedical Engineering, Hebei University of Technology, Tianjin 300130, China

**Keywords:** antibacterial, osteogenic, Mg–Fe layered double hydroxides, alkaline microenvironment, Fenton reaction

## Abstract

Osteogenic and antibacterial abilities are the permanent pursuit of titanium (Ti)-based orthopedic implants. However, it is difficult to strike the right balance between these two properties. It has been proved that an appropriate alkaline microenvironment formed by Ti modified by magnesium–aluminum layered double hydroxides (Mg–Al LDHs) could achieve the selective killing of bacteria and promote osteogenesis. However, the existence of Al induces biosafety concerns. In this study, iron (Fe), an essential trace element in the human body, was used to substitute Al, and a calcinated Mg–Fe LDH film was constructed on Ti. The results showed that a proper local alkaline environment created by the constructed film could enhance the antibacterial and osteogenic properties of the material. In addition, the introduction of Fe promoted the Fenton reaction and could produce reactive oxygen species in the infection environment, which might further strengthen the in vivo bactericidal effect.

## 1. Introduction

As the global population ages and the incidence of orthopedic diseases increases, the demand for orthopedic implants has grown rapidly [1,2]. Titanium (Ti) and its alloys have stable chemical properties and excellent biocompatibility and are often used as materials for orthopedic and dental implants [3,4]. However, most Ti-based implants fail due to the lack of antibacterial and osteogenic properties [4,5,6]. A study showed that in 337,597 knee revision surgeries, bacterial infection accounted for about 20.3% of prosthesis failure [7]. To enhance the bioactivity of conventional Ti, various nanoengineering strategies have been optimized to produce controlled nano-morphologies on Ti-based dental implants, including layer-by-layer deposition, polymer modification, and plasma sputtering. However, the effect is not satisfactory. In addition, conventional orthopedic metal implants with biologically inert surfaces are ineffective for bone formation and integration [8,9,10]. A systematic review of 11 studies reported that the occurrence of cumulative biological complications within 18 years was 43% for peri-implant mucositis and 22% for peri-implantitis. At present, the most used strategy to combat bacterial infections and poor osteogenesis is loading drugs or bio-functional metal ions (such as silver, copper, zinc, etc.) [11,12,13,14,15,16,17,18]. However, it may cause drug resistance and cytotoxicity [19]. Regulating the surface topography of implants is another widely used method, but it is only effective for bacteria and cells adhering to the sample surface. A series of stimuli-responsive implants have been developed, which can conquer implant failure by coupling with external stimuli such as light, ultrasound, magnetic fields, etc. [20,21,22,23,24,25,26]. However, the application of stimuli-responsive implants is limited by the penetration depth of the external field to the human body. The construction of biological coating is expected to achieve a balance between safety and effectiveness, which has become a hot spot of current research. 

Among the many bio-coatings, layered double hydroxides (LDHs) are widely investigated because of their high biocompatibility and ion exchange capacity [27,28]. In addition, their adjustable layer and interlayer composition and surface functionalization enable them to play different roles, such as drug delivery platforms, imaging agents, cell differentiation agents, etc. Nowadays, LDHs are widely constructed as thin films on orthopedic implants because of their excellent performance. The basic structural formula of LDHs is [M^2+^_1−x_M^3+^_x_(OH)_2_]A^n−^_x/n_•mH_2_O. M stands for metallic element and A for anion. The main structure is an octahedral hydromagnesia slate structure composed of divalent metal hydroxide, and the high metal ions replace the divalent metal ions in the original structure, thus introducing a positive charge in the plate. LDHs are biocompatible for orthopedic implants, and their anion exchange and surface adsorption capabilities can be used for the incorporation of drugs and biomolecules and controlling released at rates controlled by the microenvironment [23,29]. In addition, LDHs can be easily grown on a variety of biomedical metals and bioceramics, such as magnesium and its alloys, titanium, nickel–titanium, silicon dioxide, etc., which makes them suitable for the surface modification of biomedical implants. Recently, numerous types of LDHs have been constructed on titanium to endow the implant with antibacterial, osteogenic, and anticancer functions [30,31,32]. Among them, Mg–Al LDH has excellent antibacterial and osteogenic properties, but the Al element presents neurotoxicity, which restricts its clinical application [33]. Instead, iron (Fe) is an essential element for the human body, so Mg–Fe LDH films have attracted broad attention. In previous studies, we have verified the high biocompatibility, and osteogenic and soft tissue sealing abilities of Mg–Fe LDHs [30,34]. Unfortunately, Mg–Fe LDH films had a limited inhibitory effect on bacteria [35]. How to improve the antibacterial ability of the Mg–Fe LDH films without sacrificing their biocompatibility is still a great challenge.

In previous studies, it was found that an alkaline microenvironment is an important physical factor in bone regeneration, which has important effects on protein adsorption, cell adhesion, and cell proliferation [20,30,33,34]. In addition, the alkaline microenvironment also has certain antibacterial properties. According to previous studies, we regulated the alkalinity of the microenvironment by adjusting the temperature and time of calcination. In this study, Mg–Fe LDH modified Ti (LDH-1) was prepared and the surface alkalinity was regulated by calcination at 250 °C and 450 C to obtain new samples labeled as LDH-2 and LDH-3, respectively, as shown in Figure 1. The antibacterial and osteogenic properties of the constructed films were systemically studied. In addition, we found that the sample could be a Fenton reaction in the microenvironment containing hydrion (H^+^) and hydrogen peroxide (H_2_O_2_) to produce the hydroxyl radical (·OH), which could kill bacteria. The potential application of calcinated Mg–Fe LDH filma in orthopedic implants is discussed, which may provide new insights into the surface designing of Ti-based prostheses.

## 2. Results and Discussion

### 2.1. Sample Preparation and Characterization

Thermogravimetric (TG) analysis (Appendix A) showed that the weight loss of LDH-1 took place around 200 °C and 400 °C, which corresponded to the removal of interlayer water and lattice water, respectively. Based on the thermogravimetric result, we chose to calcinate samples at 250 °C and 450 °C to obtain LDH-2 and LDH-3, respectively. The surface morphologies of Ti, LDH-1, LDH-2, and LDH-3 samples by scanning electron microscopy (SEM) are shown in Figure 1a. The surface roughness of Ti is smoother than LDH-1, LDH-2, and LDH-3 samples (Appendix A). The roughness of Ti was about 34.30 nm. After surface modification, the surface roughness significantly increased to 176.77 nm, 106.75 nm, and 197.6 nm for LDH-1, LDH-2, and LDH-3, respectively. LDH-1, LDH-2, and LDH-3 samples show similar layered structures with a size of about 200 nm. The crystalline phase of the samples was analyzed by X-ray diffraction (XRD), as shown in Figure 1b. There is no characteristic peak of LDHs in the four groups of samples. This is because using Ti as the substrate, the thickness of the film of LDH-1 is very small. Peaks in thin films are difficult to detect compared with strong substrates. In the case of thick coating, which was prepared by prolonging the hydrothermal treatment time, the characteristic peak of LDH could be obviously observed. As shown in Appendix A, LDH-1 has a characteristic peak centered around 10° corresponding to the (0 0 3) crystalline plane of LDHs. However, no peaks are detected from LDH-2 and LDH-3 samples, which might be due to the lattice destruction caused by the calcination of LDH films at high temperatures [30]. Besides, it can be seen from Figure 1c that LDH-3 has a much higher specific surface area than LDH-1 and LDH-2. The specific surface areas of LDH-1, LDH-2, and LDH-3 are 0.6520, 1.0139, and 63.0420 m^2^/g, respectively. Furthermore, it can be seen from Appendix A that an alkaline microenvironment with different pH values can be created by calcination treatment. The local pH value of LDH-3 was the highest and was about 11.45. We believe that the change in pH on the sample surface is caused by a series of causal characteristics. To sum up, we speculate that the increase in the specific surface area of the sample film is because of the increase in the number of mesopores, which is due to the disappearance of interlayer water and lattice water during the calcination process [36]. Furthermore, the increase in specific surface area exposes a large number of hydroxyl (−OH) groups, which provided favorable conditions for the increase in local pH.

### 2.2. Antibacterial Performance of Samples

Figure 2 shows the *S. aureus* and *E. coli* colonies on the agar plates, corresponding quantitative analysis, and SEM morphology images of *S. aureus* and *E. coli* cultured on Ti, LDH-1, LDH-2, and LDH-3 samples. The number of colonies on the agar plate of the LDH-3 sample is less than that in the other groups. In addition, the quantitative results also confirm that the corresponding bacteriostatic rate of LDH-3 samples against *S. aureus* and *E. coli* is probably 53.42% and 27.59%, respectively. These results indicate that the bacteriostatic properties of LDH-3 are superior to the other three groups of samples. The SEM morphology images of *S. aureus* and *E. coli* cultured on Ti, LDH-1, LDH-2, and LDH-3 samples show that the density of colonies of *S. aureus* and *E. coli* on the LDH-3 sample is slightly lower than the other three groups of samples. On this basis, we observed the live and dead staining images of *S. aureus* and *E. coli* after 24 h (Appendix A). Some dead *S. aureus* and *E. coli* appear on the LDH-3 sample surface, with many live *S. aureus* and *E. coli* on it, indicating a weak bacteriostatic effect of LDH-3. By contrast, on the Ti, LDH-1, and LDH-2 sample surfaces, there are almost no dead *S. aureus* and *E. coli* cells, indicating good bacterial growth. The results of live and dead staining images confirm the bacteriostatic activity of the LDH-3 samples. From the above results, it can be concluded that the LDH-3 sample has certain bacterial inhibiting activity, but it does not kill the bacteria completely.

Related studies had shown that the microenvironment of bacterial infection in vivo has the characteristics of acidity and enrichment of hydrogen peroxide (H_2_O_2_) [37,38,39]. In bacterial biofilm-infected tissues, interactions between bacteria and the host result in unique biofilm microenvironments. The respiratory burst of abundant phagocytes around bacterial biofilms results in high levels of ROS (e.g., H_2_O_2_) in biofilm-infected tissues. To further explore the antibacterial performance of the LDH-3 sample, we simulated the acidic and H_2_O_2_ enrichment microenvironment of tissues with a bacterial infection. The exact concentration of hydrogen peroxide in the human body is not known, but it is generally less than 5 μM. We conducted antibacterial tests with a H_2_O_2_ concentration as low as 5 μM. As shown in Appendix A, at a concentration of 5 μM H_2_O_2_, the LDH-3 sample has a certain bacteriostatic effect. Compared with the bacteria colonies on the sample of the Ti group, the bacterial colonies on the sample of LDH-3 were reduced a lot, and the quantitative analysis results also confirmed this conclusion. To further simulate the progress of bacterial infection environment. We conducted cell proliferation experiments of Ti and LDH-3 samples under different H_2_O_2_ concentrations, as shown in Appendix A. We found that cell proliferation was not significantly inhibited at 5 and 100 μM H_2_O_2_ concentrations compared with the proliferation results at 0 μM. However, at the concentration of 200 μM, cell proliferation was inhibited to some extent. Based on ensuring the reality of the simulated environment, 200 μM of H_2_O_2_ was selected to make the research results obvious and differentiated. As shown in Figure 3a, compared with the agar plates of the Ti sample (as the control group), we could hardly observe *S. aureus* and *E. coli* colonies on the agar plates of the LDH-3 sample. The corresponding antibacterial rates of LDH-3 samples against *S. aureus* and *E. coli* are 99.17% and 93.05%, respectively. These results indicate that the antibacterial activity of the LDH-3 sample can be effectively enhanced by the infectious microenvironment. Such an excellent bactericidal effect of the LDH-3 sample might have resulted from their hydroxyl radical (·OH)-generating ability in an acidic environment enriched in H_2_O_2_. The ·OH production of Ti and LDH-3 samples were tested by a 3,3′,5,5′-tetramethylbenzidine (TMB) color reaction method in different conditions. As shown in Figure 3b, no peaks can be detected for the Ti sample in all conditions, while an apparent absorption peak appeared in the LDH-3 group when it is immersed in a solution containing H_2_O_2_ and H^+^. The corresponding quantitative analysis in Figure 3c confirms that the LDH-3 sample has a strong ·OH production capacity in an acidic (pH = 6.5) environment containing 200 μM H_2_O_2_ while generating few ·OH in other conditions. The ·OH production activity of the LDH-3 sample could be further verified by the Mott–Schotty (MS) test. As shown in Appendix A, numerous positively charged electron holes could be found in LDH-3 samples, which could be attributed to the reaction between ·OH and the Mg–Fe LDH films [40]. 

The ·OH producing ability of LDH-3 is derived from the Fenton reaction induced by the Fe element existing in the samples. The acidic microenvironment will promote the release of Fe^3+^ ions from the LDH-3 samples. These ions can catalyze the Fenton reaction in an environment rich in H^+^ and H_2_O_2_ to produce ·OH (Equations (1) and (2)) [41,42,43], while in an environment that lacks H^+^ and H_2_O_2_, the Fenton reaction can hardly proceed, resulting in little ·OH being produced from LDH-3 in the normal environment.
H_2_O_2_ + 2Fe^3+^ → 2Fe^2+^ + O_2_ + 2H^+^,(1)
H_2_O_2_ + Fe^2+^ → Fe^3+^ + (OH)^−^ + ·OH,(2)

Based on the above discussion, it can be concluded that the LDH-3 sample possesses certain bacterial inhibiting activity in the normal environment, benefiting from its alkalinity (Figure 3d). The pH of the microenvironment is an important factor in determining the viability of bacteria. Most bacteria’s metabolism depends on the H^+^ gradient across the cell membrane. Creating an alkaline microenvironment can break the transmembrane H^+^ gradient, thus inhibiting the growth of bacteria [33]. It has been verified that the higher the pH of the microenvironment, the stronger the antibacterial effect. However, an overly alkaline environment will also induce the death of osteoblasts. In this study, the local pH was controlled to 11.4, in which condition, the viability of cells will not be affected. However, the alkaline environment created by LDH-3 samples is not strong enough to kill bacteria, as illustrated by the in vitro antibacterial tests. The antibacterial ratio of LDH-3 to *S. aureus* and *E. coli* is 53.42% and 27.59%, respectively, which can hardly meet the clinical requirement. Fortunately, plenty of Fe elements exist in LDH-3 samples, promoting the Fenton reaction and producing ·OH, especially in the infectious microenvironment, characterized by high H^+^ and H_2_O_2_ concentrations. The produced ·OH can kill bacteria by causing irreversible damage to the bacterial cell wall to cause the leakage of intracellular substances [44,45]. Therefore, the LDH-3 samples show much higher bactericidal activity in the existence of H^+^ and H_2_O_2_, rendering them promising for in vivo antibacterial applications.

### 2.3. Biocompatibility of Samples

The success of cell adhesion is an important marker to test the biocompatibility of the samples. Figure 4a shows the staining images of mice bone marrow mesenchymal stem cells (BMMSCs) cultured on different sample surfaces after 4 and 24 h. After 4 h incubation, the cells on the surface of all samples show a multilateral shape, an obvious cytoskeleton, and good cell spreading. After 24 h incubation, the cells on LDH-2 and LDH-3 samples display a more extensively spread cytoplasm, while the cells on Ti and LDH-1 samples exhibit a slightly round shape with smaller spreading areas. However, the difference between the four samples is not significant. Quantitative results show that the numbers of attached BMMSCs on LDH-2 and LDH-3 are larger than Ti after incubation for 4 h, whereas LDH-1 processes a slightly larger amount of BMMSCs latching onto its surface than Ti with no significant difference. After 24 h, the number of cells on Ti surface is greater than LDH films. These results indicate that the calcined Mg–Fe LDH film did not significantly inhibit the spread and adhesion of BMMSCs. On this basis, we investigated the effect of different samples on the viability of BMMSCs. The cellular viability was tested by live/dead cell staining. It can be seen from Figure 4b that all samples are covered by live cells (green). Only a small number of dead cells stained by propidium iodide (red) are presented on the surface of Ti, LDH-1, and LDH-2. In conclusion, the samples do not show obvious toxic side effects on cells.

Figure 5a shows the proliferation of BMMSCs cultured on the surfaces of various samples for 1, 2, 3, and 5 days. With the increase in incubation time, the proliferation rate of cells on the surface increases in a time-related manner. The cell proliferation rate on the surface of the Ti sample is higher than LDH-1 and LDH-2 after culturing for 1, 2, 3 and 5 days, while that is lower than LDH-3. The trend is as follows: LDH-3 > Ti > LDH-2 > LDH-1. However, these results indicate that the LDH-3 sample could promote cell proliferation. Figure 5b presents the expression levels of osteogenic genes of BMMSCs cultured on different samples detected by qRT-PCR after 3 and 7 days. LDH-2 group presents the highest expression of Runx2, Oxterix, COL1, and OCN gene mRNA among the four groups after 3 days. Additionally, LDH-3 sample also promotes the expression of COL1 after 3 days compared to the Ti sample. Subsequently, after 7 days of incubation, LDH-1, LDH-2, and LDH-3 significantly promote the expression of the COL1 gene in BMMSCs. Furthermore, there is no significant difference in the expression of other genes among the four groups of samples. It is worth noting that the expressions of COL1 in BMMSCs on LDH-3 samples far exceeded the corresponding levels of cells on other samples. The above results indicate that the LDH-3 sample not only promoted the proliferation of BMMSCs but also enhanced their osteogenesis differentiation.

According to previous research, the ability of samples to promote osteogenic differentiation of BMMSCs mainly comes from the alkaline microenvironment created by the samples [33,34,46,47,48]. It is reported that a weak alkaline environment could promote the activation of integrin receptors, which stimulates the phosphorylation of FAK and further upregulates the expression of osteogenic genes such as ALP and Runx2 [49]. In addition, Mg^2+^ release might be another factor promoting the osteogenic differentiation of BMMSCs cultured on the prepared samples. It had been verified that Mg^2+^ can activate several relevant osteogenic differentiation signaling pathways, such as PI3K-AKT and MAPK, which further promote the transcription of osteogenic-related BMP-4 gene and ultimately promote the osteogenic differentiation of cells [47]. As illustrated in Figure 5c, the prepared LDH-3 samples could not only generate a locally alkaline microenvironment but also release Mg^2+^ (Figure 5c), so they are beneficial to the osteogenic differentiation of BMMSCs.

## 3. Materials and Methods

### 3.1. Sample Preparation

The Ti (TA3) sheet (10 mm × 10 mm × 1 mm) was ultrasonically cleaned with an acid solution (HF:HNO_3_:H_2_O = 1:5:4) successively, and the sample was labeled as Ti. LDH-1 was constructed by the hydrothermal method at 105 °C for 16 h in an electric heating drying box (DL-101-2, Tianjin Tongda Experimental Furnace Factory, Tianjin, China). The total concentration of metal ions in the hydrothermal reaction solution was 80 mM, and the concentration ratio of magnesium ions to iron ions was 3:1. The iron and magnesium salts used were iron chloride hexahydrate (AR, Aladdin, Shanghai, China) and magnesium chloride hexahydrate (AR, Aladdin, Shanghai, China), respectively. The LDH-1 samples were calcined in a muffle furnace (SG-G0316, Tianjin Zhonghuan Experimental Electric Furnace Co., Ltd., Tianjin, China) at 250 °C and 450 °C, and the heating rate was 10 °C/min. After rising to the selected temperature, the holding time was 2 h, and the samples were naturally cooled to room temperature. The obtained samples were numbered LDH-2 and LDH-3, respectively.

### 3.2. Surface Morphology and Crystalline Phase Characterization

SEM (S-4800, Tokyo, Japan) was used to observe the surface morphology of the samples at different magnifications. XRD (Bruker D8, Karlsruhe, Germany) was used to analyze the phase composition of the sample surface, with a scanning range of 5~90°. TG analysis (SDT Q-600, Shanghai, China) was used to measure weight changes associated with transitions and reactions inside materials. A physical-chemical adsorption analyzer (BET, autosorb iQ, FLA, USA) was used to measure the specific surface area of the material. Atomic force microscopy (AFM, 5500, Agilent Technologies, Shanghai, China) was used to observe the surface roughness of the sample. 

### 3.3. pH Measurement

First, plates were completely submerged in deionized water (5 mL per sample) for 24 h at room temperature to determine the capacity of LDH modified plates to adjust the solution pH value. Then, the ability of samples on changing the microenvironment alkalinity was tested. Briefly, 4 mg powders were scraped from each group and powders were submerged in 5 mL of deionized water for 24 h at room temperature. A pH meter tested the pH changes of deionized water.

### 3.4. Electrochemical Performance

An electrochemical workstation (CHI760C, Chenhua, Shanghai, China) was used to test the MS of the prepared coatings. The semiconducting properties of different samples were studied by MS analysis, which was performed at underpotentials ranging from −2 V to 2 V vs. SCE with an interval of 0.05 V at a frequency of 1000 Hz.

### 3.5. Hydroxy Radicals Detection

The ·OH were detected by the TMB color reaction method. First, 20 μL of 9.79 mM H_2_O_2_ was added to 1 mL PBS (0.01 M pH = 6.5, Solarbio, Beijing, China) and mixed well. Then, 1 mL of the mixed PBS and 20 μL TMB were added to the surface of the sample. Subsequently, after 15 min at room temperature, the absorbance at 652 nm was measured by ultraviolet spectrophotometry (UV, UV-6100, Shanghai, China).

### 3.6. Bacteria Culture

The materials were placed onto an ultra-clean table for ultraviolet light sterilization for 2 h. The bacterial solution was further diluted with Luria-Bertani (LB, 6g Tryptone (pH = 7.3 ± 0.2, amino nitrogen 3.7% w/w, sodium chloride 0.4% w/w, OXOID, UK), 1.8 g beef extract powder (HB8272, Haibo Biotechnology Co., Ltd. Qingdao, China) and 3 g sodium chloride (99.5% Biosharp, Beijing, China) into 600 mL deionized water for high temperature sterilization) until the optical density (OD) was 0.1 (corresponding to 10^7^ CFU/mL for the bacterial solution). Then, the bacterial solution with OD = 0.1 was diluted 10 times (10^6^ CFU/mL) with PBS (0.01 M pH = 7.2–7.4, Solarbio, Beijing, China). Then, 30 μL of the bacterial solution was added to each sample and placed into an incubator at 37 °C for co-culture for 24 h. Bacteria on the co-culture samples were dispersed and diluted with sterile PBS (0.01 M pH = 7.2–7.4, Solarbio, Beijing, China) (10-fold, 100-fold, 1000-fold), and three parallel samples were set for each fold. For each parallel sample, 100 μL diluted bacterial suspensions were added to the agar culture plate and then coated with a coater. The agar plate coated with bacteria was incubated upside down for 16–18 h in an incubator at 37 °C. Then, the agar plates were photographed. The antibacterial ratio was statistically calculated.
$$Antibacterial\;rate\;C\%=\frac{C_{0}-C_{1}}{C_{0}}\times 100\%$$

*C*_0_ is the number of colonies in the blank group, and *C*_1_ is the number of colonies in the experimental group.

### 3.7. Morphology of Bacteria

After incubation for 24 h, 0.5 mL of glutaraldehyde solution (2.5 vol%) was added to each well to immobilize the bacteria for 6 h. Then, the bacteria were dehydrated by using water and ethanol mixtures with volume fractions (ethanol vol%) of 30, 50, 75, 90, 95, and 100%, respectively. After dehydration, the samples were dried at room temperature. Then, SEM was used to observe the bacteria after spraying gold.

### 3.8. Bacteria Live/Dead Staining

The bacteria on the sample surfaces were stained using the Live/Dead BacLight kit (L13152, Thermo Fisher Scientific, Waltham, MA, USA). Briefly, after being cultured for 24 h, the bacteria were rinsed twice with 0.5 mL of PBS (0.01 M pH = 7.2–7.4, Solarbio, Beijing, China), 0.5 mL of dye was added to each sample and then stained for 15 min at room temperature. Afterward, the samples were rinsed with 0.5 mL of PBS and then observed with a fluorescence microscope.

### 3.9. Bacteria Culture under the Acidic and H_2_O_2_ Environment

The materials were placed onto an ultra-clean table for ultraviolet light sterilization for 2 h. The bacterial solution was further diluted with LB until the OD was 0.1 (corresponding to 10^7^ CFU/mL for the bacterial solution). Then, the bacterial solution with OD = 0.1 was diluted 10 times (10^6^ CFU/mL) with PBS (0.01 M pH = 6.5, LeiGen Biotechnology Co., LTD, Huaibei, China). Subsequently, 1 mL bacterial solution (pH = 6.5) containing H_2_O_2_ (5 μM and 200 μM) was added and mixed well. Then, 30 μL of the bacterial solution was added to each sample and placed into an incubator at 37 °C for co-culture for 24 h. Bacteria on the co-culture samples were dispersed and diluted with sterile PBS (0.01 M pH = 7.2–7.4, Solarbio, Beijing, China) (10-fold, 100-fold, 1000-fold), and three parallel samples were set for each fold. For each parallel sample, 100 μL diluted bacterial suspensions were added to the agar culture plate and then coated with a coater. The agar plate coated with bacteria was incubated upside down for 16–18 h in an incubator at 37 °C. Then, the agar plates were photographed. The antibacterial ratio was statistically calculated.
$$Antibacterial\;rate\;C\%=\frac{C_{0}-C_{1}}{C_{0}}\times 100\%$$

*C*_0_ is the number of colonies in the blank group, and *C*_1_ is the number of colonies in the experimental group.

### 3.10. Cell Initial Adhesion

Samples were placed in 24-well plates. BMMSCs (5 × 10^4^ cells per well) were seeded on samples and cultured for 4 and 24 h. Samples were washed twice and fixed with paraformaldehyde (PFA, 4%) for 15 min. Then, BMMSCs were permeabilized with 0.1% Triton X-100 solution (Sigma-Aldrich, St. Louis, MO, USA) for 5 min and incubated with 1% bovine serum albumin blocking solution (BSA; Sigma-Aldrich, MO, USA) for 25 min at room temperature. Later, cells were stained by TRITC phalloidin (Solarbio, Beijing, China) to visualize filamentous actin (F-actin) for 30 min and mounted with a mounting medium with DAPI (Vector, Torrance, CA, USA) for 20 min, consecutively. Lastly, a confocal laser-scanning microscope (CLSM, Olympus, Toyko, Japan) enabled the collection of fluorescence images.

### 3.11. Cell Live/Dead Staining

Samples were placed in 24-well plates. BMMSCs (5 × 10^4^ cells per well) were seeded on samples and cultured for 4 and 24 h. Then, calcium-AM and propidium iodide (PI) were diluted in PBS with a final concentration of 2 and 5 μM, respectively, and a 100 μL diluted solution was added to each sample. After being cultured for another 15 min, the stained cells were observed using a fluorescence microscope.

### 3.12. Cell Proliferation

Cell proliferation was evaluated using AlamarBlue™ assay (Thermo Fisher Scientific Inc., MA, USA). BMMSCs (5 × 10^4^ cells per well) were seeded on samples and cultured for 1, 2, 3, and 5 d. Then, samples were washed twice and incubated in a 500 μL medium containing 10% AlamarBlue for 2 h. The fluorescence intensity was determined using excitation and emission wavelengths of 560 and 590 nm, respectively.

### 3.13. Cell Proliferation under the H_2_O_2_ Environment

Cell proliferation was evaluated using AlamarBlue™ assay (Thermo Fisher Scientific Inc, MA, USA). BMMSCs (5 × 10^4^ cells per well) were seeded on Ti and LDH-3 samples and cultured under 0, 5, 100, and 200 μM H_2_O_2_ for 1 d. Then, samples were washed twice and incubated in a 500 μL medium containing 10% AlamarBlue for 2 h. The fluorescence intensity was determined using excitation and emission wavelengths of 560 and 590 nm, respectively.

### 3.14. Gene Expression by Quantitative Real-Time Polymerase Chain Reaction (qRT-PCR) Analysis

The qRT-PCR assay was conducted to investigate the relative mRNA expression of Runx2, Osterix, ALP, COL1, and OCN. BMMSCs (5 × 10^4^ cells per well) were seeded on sample surfaces and cultured for 3 and 7 d. The total RNA was isolated using TRIzol reagent (Thermo Fisher Scientific Inc., MA, USA). First-strand cDNA was synthesized from 1 to 2 μg total RNA using the PrimeScriptRT MasterMix (TaKaRa, Osaka, Japan), followed by qRT-PCR carried out with SYBR Select Master Mix (Bio-Rad, Hercules, CA, USA) and specific oligonucleotide primers. 

### 3.15. Statistical Analysis

Statistical analysis was assessed by using GraphPad Prism statistical software package. Statistically significant differences (P) were analyzed by t-tests, one-way analysis of variance (ANOVA), two-way ANOVA, and Tukey’s multiple comparison tests. *p* < 0.05 is marked by “*”, *p* < 0.01 is marked by “**”, *p* < 0.001 is marked by “***”, and *p* < 0.0001 is marked by “****”.

## 4. Conclusions

In this work, we propose a novel strategy to modulate the surface alkalinity of Ti to improve its properties. The Mg–Fe LDH film was grown on the Ti surface by hydrothermal treatment. The pH of the local microenvironment of the film rises. Antibacterial experiments in vitro showed that Mg–Fe LDH films had certain bacteriostatic activity after calcination at high temperatures, though it cannot kill all bacteria, However, in the simulated in vivo bacterial infection environment, we found that the LDH-3 sample could promote the Fenton reaction to produce ·OH owing to the existence of the Fe element, and its antibacterial rate was enhanced to above 90%. In addition, the LDH-3 sample showed good cytocompatibility and could promote the osteogenic differentiation of BMMSCs, which is promising for the clinical application of hard tissue implants.

## Data Availability

The authors declare that all data supporting the findings are available within the paper or are available from the authors upon request.

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
