# Peer review of "Enhanced Bactericidal Effect of Calcinated Mg–Fe Layered Double Hydroxide Films Driven by the Fenton Reaction"

_ijms, 2022, doi:10.3390/ijms24010272_

Round 1
Reviewer 1 Report
This study investigated the osteogenic and antibacterial properties of calcinated Mg-Fe layered double hydroxides films developed on titanium substrates via hydrothermal methods and heat treatments. Overall, the experiments have been performed in details and results compliment the hypothesis of this study. Below are my comments for editor’s and author’s consideration:
1. In sample preparation authors have mentioned that “The Ti sheet (10 mm×10 mm×1 mm) was ultrasonically cleaned with an acid solution (HF:HNO3:H2O=1:5:4) successively, and the sample was labelled as Ti”. What was the grade of titanium used in this study? Is it commercially pure titanium or a Ti alloy? Did authors check the impurities in the Ti sheet before further surface modification processing via ICP-AES?
2. Mention sources of magnesium and iron ions used in this study?
3. Provide model # of muffle furnace and other equipment used in this study.
4. Similarly, for solutions/chemical used in antibacterial tests, sources and purity levels should be mentioned.
5. In results section, authors mentioned that “The surface of Ti is rough and uneven”. This statement sounds vague. What was the actual roughness of the Ti sheet before and after the surface modification? Can be measured using AFM or a profilometer.
6. Authors mentioned that “samples show similar layered structures with a size of about 200 nm”. How did they measured that to be 200 nm. This information is missing in the manuscript.
7. In XRD spectra (Fig. 1b), peaks are not clear in the range of 6-27 degrees. Remove the noise from the XRD spectrum for each scanned sample and replot.
8. Include scale bars in 24 h results in Fig. 4a.
9. In conclusion, authors mentioned that “we propose a novel strategy to modulate the surface alkalinity of Ti to conquer implant infection and poor osseointegration”. Does Ti-based implants show poor osseointegration? What about Ti-implants being used successfully in dental applications? This statement should be revised as there was no quantitative comparison made by existing Ti-based implants. Moreover, no in vivo results were presented to prove this claim.
Author Response
We would like to thank you for your time in closely viewing our manuscript, and also thank you for giving us constructive suggestions which would help us to improve the quality of the paper. Here are our responses to your questions. Please see the attachment for details.

Reviewer 2 Report
Mg-Al LDHs modified Ti is investigated in this manuscript. Potential antibacterial and osteogenic properties have been examined. It is suggested that the following concerns should be addressed before publication.
The biggest concern is that the claimed “simulated in vivo bacterial infection environment” needs to be validated. H2O2 concentration used in this study is 100 μΜ. In fact, according to the methods section 3.8, the concentration should be 200 μM. Please double check the number. However, according to the reviewer’s knowledge, the physiological H2O2 concentration is less than 5 μM. Why is this physiologically irrelevant concentration claimed as a simulated in vivo condition? In addition, almost no human cell can stand this high H2O2 concentration either. Will this antibacterial finding exist in real applications?
It is claimed that the surface pH of LDH-3 is 11.4. How is this number determined? Is this in vitro or in vivo? Related text should be added to the methods section.
Please explain Figure 3d. What do the two lines of blocks mean, respectively?
Figure 4a, vertical axis, what does Cell Numbers actually means? Cell number per view or something?
It is noticed that two-way ANOVA was used according to 3.13. Which data were analyzed by two-way ANOVA? Also it looks like not all the significant differences are labeled. Any explanations?
Line 239, there is no Figure 6c.
Author Response

(The authors gave the same response as above.)

Round 2
Reviewer 2 Report
All previous concerns were carefully addressed. Great improvement has been made compared with the previous version.